# Insights into the Evolution of Spermatogenesis-Related Ubiquitin–Proteasome System Genes in Abdominal Testicular Laurasiatherians

**DOI:** 10.3390/genes12111780

**Published:** 2021-11-10

**Authors:** Xiaoyue Ding, Li Cao, Yu Zheng, Xu Zhou, Xiaofang He, Shixia Xu, Wenhua Ren

**Affiliations:** Jiangsu Key Laboratory for Biodiversity and Biotechnology, College of Life Sciences, Nanjing Normal University, Nanjing 210023, China; 181201004@njnu.edu.cn (X.D.); 201202081@njnu.edu.cn (L.C.); 191202080@njnu.edu.cn (Y.Z.); 191202030@njnu.edu.cn (X.Z.); 201202050@njnu.edu.cn (X.H.); xushixia78@163.com (S.X.)

**Keywords:** laurasiatherian, spermatogenesis, ubiquitin–proteasome system gene, positive selection, rapid evolution, molecular convergence, functional convergence

## Abstract

During embryonic development in mammals, the testicles generally descend into the scrotum, making the testicular temperature 2–4 °C lower than the core temperature via heat exchange and clearance, and thus more beneficial for normal spermatogenesis. Failure to descend, known as cryptorchidism, carries a series of risks such as infertility and testicular cancer. However, some mammals have evolved abdominal testes while maintaining healthy reproduction. To explore the underlying molecular mechanism, we conducted comparative genomic analyses and functional assays on the spermatogenesis-related ubiquitin–proteasome system (UPS) genes essential to sperm formation in representative laurasiatherians. Here, positive selection and rapid evolution of spermatogenesis-related UPS genes were identified in the abdominal testicular laurasiatherians. Moreover, potential convergent amino acids were found between distantly related species with similar abdominal testicles and functional analyses showed RNF8 (V437I) in abdominal testicular species (437I) has a stronger ubiquitination ability, which suggests that the mammals with abdominal testes might exhibit enhanced sperm cell histone clearance to maintain sperm formation. This evidence implies that, in response to “cryptorchidism injury”, spermatogenesis-related UPS genes in the abdominal testicular species might have undergone adaptive evolution to stabilize sperm formation. Thus, our study could provide some novel insights into the reproductive adaptation in abdominal testicular mammals.

## 1. Introduction

The testis, the organ in which spermatogenesis begins, is important for male reproduction. During the embryonic development of mammals, the testes usually descend from the urogenital ridge through the abdomen and into the scrotum [1]. Two main processes of the scrotum make the testicular temperature 2–4 °C lower than the core temperature: heat exchange with the environment through the scrotum skin and heat clearance by blood flow through the pampiniform plexus [2]. In addition, spermatogenesis, the most basic biological process of male reproduction, is vulnerable to temperature [3]. Thus, cryptorchidism—the process in which the testes do not descend into the scrotum and instead remain in the abdominal cavity during development—exposes the testes to the higher temperature of the abdominal cavity; this can hamper spermatogenesis, causing male infertility and other severe secondary testicular diseases [4,5].

Most mammals, such as primates, have testes located in the scrotum outside the abdominal cavity, i.e., scrotal testicular mammals [6]. For them, the testes failing to fall leads to cryptorchidism [7]. However, not all mammals have the structure of the scrotum, so the testicles do not descend to the scrotum. The abdominal testicular mammals have testes remaining in the urogenital ridge near the kidneys, showing no testicular descent (e.g., most afrotherians) or just descending into the lower abdomen, like dolphins and seals [6,8,9]. However, these abdominal testicular mammals can undergo normal spermatogenesis and maintain healthy reproduction functions, unlike scrotal testicular mammals that suffer from “cryptorchidism injury”. Some mammals with “healthy cryptorchidism” have been reported to have anatomical adaptations that maintain healthy reproduction, e.g., seals can use the cooled surface blood to directly cool their reproductive organs [10]. The molecular mechanism underlying this, however, is still unclear. Interestingly, in laurasiatherians, many closely related species have different testicular phenotypes. For example, in Cetartiodactyla, Ruminantia and Camelidae are scrotal testicular mammals, but cetaceans and Hippopotamidae are abdominal testicular mammals; in Perissodactyla, Equidae are scrotal testicular mammals, while Rhinocerotidae and Tapiridae are abdominal testicular mammals [6]. For this reason, laurasiatherians are an ideal model for studying the mechanism driving the evolution of this so called “healthy cryptorchidism”.

Spermatogenesis is a complex process of cell division and differentiation. It requires precise expressions of structural proteins and enzymes, which are affected not only by the regulation of gene transcription and translation, but also by the degradation of various proteins. The ubiquitin–proteasome system (UPS), a proteolytic system, is necessary for various steps of mammalian spermatogenesis and fertilization [11,12]. Ubiquitin-mediated proteolysis is involved in the establishment of both spermatogonial stem cells and differentiating spermatogonia from gonocytes [13]. For instance, ubiquitination of histones in the testes leads to histones being replaced by protamines, which are essential for the condensation of chromatin and spermatogenesis [14]. Moreover, ubiquitination plays important roles in several key processes during meiosis such as genetic recombination and sex chromosome silencing [15,16].

UPS components include ubiquitin, ubiquitin priming enzyme, proteasome and deubiquitinated protease. Ubiquitin promoter enzymes include ubiquitin activating enzyme (E1), ubiquitin conjugating enzyme (E2), and ubiquitin ligase (E3), which are responsible for activating ubiquitin and binding it to the substrate protein to be degraded, adding to the target protein on the ubiquitin label, that is, ubiquitination [17]; proteasomes can recognize and degrade target proteins that have been ubiquitinated [18]; deubiquitinating enzymes (DUBs) are responsible for dissociating the ubiquitin chain from the target protein so that the ubiquitin can be recycled into the cytoplasm [19]. Numerous functions of the UPS genes—including E1, E2, E3, proteases and DUBs in mammalian spermiogenesis—have been proposed. For instance, *UBA1*/*UBE1*, an E1, initiates the cascade of UPS and is involved in male reproduction via spermatogenesis [20]; *UBE2A*/*HR6A*, an E2, is required for the maintenance of X chromosome silencing in spermatocytes and spermatids [21]; *RNF8*, an E3, participates in sperm cell nucleosome remodeling through the ubiquitination of histones [22,23]; *PSMA8*, a testis-specific 20S core proteasome subunit, degrades pro-meiotic I protein during spermatogenesis [24]; *UCHL3*, a DUB, plays a role in regulating germ cell apoptosis and the differentiation of spermatocytes into sperm cells [25,26]. However, the functions of these genes in abdominal testicular species are poorly understood.

In the present study, we performed comparative genomic analyses of spermatogenesis-related UPS genes in representative laurasiatherians and further functional analyses on RNF8. Positive selection, accelerated evolution and convergent evolution of spermatogenesis-related UPS genes, together with functional evidence of RNF8, provided some new insights into the reproductive adaptation in abdominal testicular mammals.

## 2. Materials and Methods

### 2.1. Selected Species

A total of 26 laurasiatherians representing two phenotypic classifications (2 eulippotyphla, 6 carnivora, 2 perissodactyla, 11 cetartiodactyla, and 5 chiroptera) of abdominal testicular and scrotal testicular were selected in our study. This included 12 abdominal testicular species—*Tursiops truncatus*, *Orcinus orca*, *Delphinapterus leucas*, *Physeter catodon*, *Balaenoptera acutorostrata*, *Ceratotherium simum*, *Neomonachus schauinslandi*, *Odobenus rosmarus*, *Pteropus alecto*, *Pteropus vampyrus*, *Condylura cristata Erinaceus europaeus*— and 14 scrotal testicular species—*Bos taurus*, *Ovis aries*, *Capra hircus*, *Sus scrofa*, *Vicugna pacos*, *Camelus ferus*, *Equus caballus*, *Canis lupus*, *Mustela putorius*, *Ailuropoda melanoleuca*, *Felis catus*, *Myotis davidii*, *Myotis lucifugus* and *Eptesicus fuscus.* There were 10 non-laurasiatherian mammals: five abdominal testicular species—*Loxodonta africana* (Proboscidea), *Trichechus manatus* (Sirenia), *Orycteropus afer* (Tubulidentata), *Dasypus novemcinctus* (Cingulata) and *Ornithorhynchus anatinus* (Monotremata)—and five scrotal testicular species—*Homo sapiens* (Primates), *Rattus norvegicus* (Rodentia), *Oryctolagus cuniculus* (Lagomorpha), *Tupaia chinensis* (Scandentia) and *Monodelphis domestica* (Ameridelphia). 

### 2.2. Candidate Genes and Sequence Acquisition

We screened a total of 25 spermatogenesis-associated UPS candidate genes (Appendix A). The whole coding regions of these genes were mostly downloaded from the National Center for Biotechnology Information (NCBI https://www.ncbi.nlm.nih.gov/ (accessed on 1 October 2021)), the accession numbers of which are listed in the Appendix A. For the sequences that were not available from the online database, we used our laboratory’s script to extract each exon of the candidate gene from the genome downloaded by NCBI with the well-annotated gene sequences of kinship as queries. Then, we connected the exons into the whole coding sequences according to the known coding sequences of species. The nucleotide and amino acid sequences of each gene were aligned using Muscle in MEGA 6.0 [27] and trimmed manually.

### 2.3. Selective Pressure Analysis

For the molecular evolution analysis, selective pressure and evolution rate were calculated by comparing the ratio of nonsynonymous substitution (*d*_N_) to synonymous substitution (*d*_S_), known as ω, based on phylogenetic methods. ω > 1, ω = 1 and ω < 1 represent genes subject to positive selection, neutral selection and purifying selection, respectively. The codon-based maximum likelihood models implemented in the CODEML program in PAML 4.7a [28] were employed to estimate the rates of synonymous (*d*_S_) and nonsynonymous substitutions (*d*_N_), as well as the *d*_N_/*d*_S_ ratio (ω).

To detect whether the abdominal testicular lineages are subject to positive selection, we used branch models including the free-ratio model and two-ratio model [29,30], and the branch-site model implemented in CODEML [28]. The free-ratio model allowed each evolutionary branch to have its own ω. The two-ratio model and branch-site model required the foreground branches (lineages tested to be under positive selection) and background branches (rest of the lineages) to be defined a priori. For each gene, each branch of the abdominal testicular lineages was treated as a foreground branch, whereas the remaining branches were treated as a background branch. The two-ratio model was used as an alternative hypothesis for the branch model, which allowed for different ω values between the foreground and background branches. As the corresponding null hypothesis, the one-ratio model showed all evolutionary branches to have the same ω value. The alternative hypothesis of the branch-site model in this study allowed each codon of the foreground branch (non-scrotal testicular species) to have its own ω value, and it allowed its ω to be greater than 1 (positive selection model: 0 < ω_0_ < 1, ω_1_ =1 and ω_2_ ≥ 1); the null hypothesis (neutral model: 0 < ω_0_ < 1, ω_1_ = 1 and ω_2_ = 1) did not allow positive selection to occur. Then, the likelihood ratio test (LRT) with a χ^2^ distribution was used to determine which models were statistically different from the null model at a threshold of *p* < 0.05. Moreover, the *p*-values of all genes were multiple-calibrated by FDR (false discovery rate) using the method of Benjamini–Hochberg [31]. For positive selection sites, Bayes empirical Bayes (BEB) analysis was used to determine sites under positive selection with posterior probabilities ≥ 0.8 [32].

To explore whether there was a significant difference in the selection pressure of spermatogenesis-associated UPS genes between abdominal testicular and scrotal testicular species, and to detect rapid evolution genes in abdominal testicular species, the nested branch model (two-ratio model) was used to calculate the selection pressure. We defined rapid evolution genes in abdominal testicular species to be genes for which the evolution rate of the foreground branch (abdominal testicular species) was greater than that of the foreground branches (scrotal testicular species), after LRT test and FDR correction, and the *p*-adjusted value was <0.05.

### 2.4. Labeling Positive Selection Sites on the Three-Dimensional Structure of Proteins

To highlight the importance of positively selected sites in terms of protein function, we mapped the sites onto the 3D structure of proteins. First, UCHL3 and PSMA8 sequences from the bottlenose dolphin were used to predict the 3D structure by I-TASSER (https://zhanglab.ccmb.med.umich.edu/services/ (accessed on 1 October 2021)). Then, we used the UniProt website (http://www.UniProt.org/ (accessed on 1 October 2021)). to view the important functional domains of each gene. Finally, the positively selective sites and functional domains were annotated in the obtained 3D structure using EzMol (http://www.sbg.bio.ic.ac.uk/ezmol/ (accessed on 1 October 2021)). and Adobe illustrator. 

### 2.5. Identification of Convergent Amino Acids among Abdominal Testicular Laurasiatherians

To explore whether molecular convergence occurred among the abdominal testicular species, we used the MEGA and Fasparser softwares [33] to compare the amino acid sequences of spermatogenesis-related UPS genes. We used amino acid sites that were shared among multiple abdominal testicular species but differed from most scrotal testicular species; such amino acids were defined as convergent in abdominal testicular laurasiatherians. Moreover, we selected some genes and expanded the laurasiatherians dataset to all mammals to further verify these putatively convergent amino acids. 

### 2.6. Functional Assays of RNF8

#### 2.6.1. Plasmid Construction and Transient Transfection

The *RNF8* gene of the bottlenose dolphin and *H2A* gene of *H. sapiens* were optimized and synthesized by Shanghai Generay Biotech Co., Ltd. (Shanghai, China). Then, we cloned *RNF8* from the bottlenose dolphin into a Myc-tagged pcDNA3.1 V5-His C plasmid and cloned *H2A* of *H. sapiens* into a Flag-tagged pcDNA3.1 V5-His C plasmid. In addition, we constructed a mutant RNF8 plasmid by changing Ile into Val at position 437 in RNF8 of the bottlenose dolphin. Mutant *RNF8* of the bottlenose dolphin (RNF8 I437V) was generated by a Q5 Site-Directed Mutagenesis Kit. The primers are as follows: forward primer: CTGCATTTCTGAGTGGATGAAGCGGAAGGTGGAGTGCCCTATTTGCCGCAAGGACATTA; reverse primer: TAATGTCCTTGCGGCAAATAGGGCACTCCACCTTCCGCTTCATCCACTCAGAAATGCAG. All genes were verified by sequencing. *RNF8*, mutant *RNF8* and *H2A* recombinant plasmids were each transfected into a different set of HEK293T cells with Lipofectamine 3000 transfection reagent (Life Technology).

#### 2.6.2. Cell Culture, Protein Extraction and Western Blot Analysis

HEK293T cell lines were cultured in DMEM medium containing 10% fetal bovine serum (FBS, WISENT) and 1% penicillin/streptomycin (P/S) at 37 °C and 5% CO_2_. Nucleoproteins were extracted through Boster’s subcellular structure nuclear and cytoplasmic protein extraction kit. This kit used a low osmotic pressure condition to fully expand the cells and then destroy their cell membrane to release the cytoplasmic protein; the cells were then centrifuged to obtain the nucleus precipitation. Finally, the nucleoprotein was extracted with a high-salt nucleoprotein extraction reagent. The prepared nucleoprotein samples were boiled at 95 °C for 5 min and then separated by SDS/PAGE. Then, they were transferred to nitrocellulose membrane and s-blocked with 5% skimmed milk powder. Proteins were detected with the following antibodies: mouse anti-Flag (Affinity), 1:3000; rabbit C-MYC-tag (Affinity), 1:3000; rabbit anti-H4 (Proteintech), 1:800; and goat anti-rabbit/mouse IgG (H + L) HRP (Proteintech), 1:2000. The blots for detecting proteins were semi-quantified by NIH Image J software (National Institutes of Health, Bethesda, MD, USA).

#### 2.6.3. Statistical Analysis

All the experiments in the functional assays of RNF8 were performed with at least three replicates. All the data were expressed as the mean ± SEM. The differences between groups were examined by a two-tailed Student *t*-test.

## 3. Results

### 3.1. Positive Selection in Abdominal Testicular Mammals

Twenty-five instances of positive selection were identified in 15 genes from the laurasiatherian dataset using the free-ratio model. Of these, 13 signs of positive selection were found in the abdominal testicular lineages, while only four were found in the scrotal testicular lineages; five signs of positive selection were identified in three laurasiatherian genes using the two-ratio model, all of which were found in the abdominal testicular lineages (Appendix A). This suggests that there were far more positive selection signals detected in the abdominal testicular lineages than in the scrotal testicular lineages. Moreover, we specifically detected positive selection in six genes (*UCHL3*, *PSMA8*, *USP14*, *MARCH7*, *USP2* and *USP14*) in the abdominal testicular branches using the free-ratio model, three of which (*UCHL3*, *PSMA8* and *USP14*) underwent positive selection according to the two-ratio model with the adjusted *p* value corrected by FDR < 0.05 (Figure 1; Appendix A). In other words, ω (the ratio of nonsynonymous substitution to synonymous substitution) > 1 was restricted to the abdominal testicular branches for these genes, e.g., the terminal branch of the sperm whale (*P. catodon*) for *UCHL3*; the LCA branch of the killer whale (*O. orca*) and bottlenose dolphin (*T. truncatus*), and LCA branch of the killer whale (*O. orca*) and white whale (*D. leucas*) for *PSMA8* (Figure 1; Appendix A). Notably, when *p* value was tested by LRT (the likelihood ratio test) < 0.05, *UCHL3* and *PSMA8* were specifically detected positive selection on the abdominal testicular branches by the branch-site model; *MARCH7* was also specifically detected positive selection on the abdominal testicular branches by the two-ratio model; more positive selection signals were specifically detected on the abdominal testicular branches (Appendix A). 

### 3.2. Rapid Evolution Rates in the Abdominal Testicular Lineages

Using the branch model (two-ratio model) in PAML, six genes were identified as having significantly higher ω values in the abdominal testicular species than the scrotal species, *p* value tested by LRT < 0.05 (Appendix A). After FDR correction, there were still four genes (*UBE2A*, *UCHL3*, *HERC4* and *PSME4*) identified in the abdominal testicular species with a higher ω value than the scrotal species. The ω values of *UBE2A* and *UCHL3* were up to over four-fold higher in the abdominal testicular species than in the scrotal testicular species (Figure 2; Appendix A). Only one gene (*RFP*) had a significantly higher ω value in the scrotal species than in the abdominal testicular species (Appendix A). No gene had the same ω value in the two testicular phenotypic species.

### 3.3. Potential Molecular Convergence between Abdominal Testicular Species

We identified eight potential convergent amino acids for eight genes (RNF8 V437I, MARCH7 V314I, UBA7 Q/K713R/E, UCHL3 V64I, PSMA8 N187S, MARCH10 P132S, UBA1 Y323F and RAD18 S440P). For three convergent amino acids (RNF8 V437I, MARCH7 V314I and UCHL3 V64I), most abdominal testicular species were Ile and most scrotal testicular species were Val (Figure 3A). Then, for RNF8 and MARCH7, we verified representative, non-laurasiatherian mammals and found that, at position 437 of RNF8 and position 314 of MARCH7, most abdominal testicular species still converged to Ile (Figure 3B,C).

### 3.4. Functional Convergence of the RNF8 among Abdominal Testicular Species

The RNF8 V437I convergence amino acid was identified in laurasiatherians with abdominal testicles and verified by representative, non-laurasiatherian mammals. The amino acid residue at location 437 of RNF8 in the abdominal testicular species was Ile in all the studied species, except for seals (*N. schauinslandi*), walruses (*O. rosmarus*) and duckmoles (*O. anatinus*) (Figure 3A,B and Figure 4A). Moreover, the V437I convergence amino acid was located at the RING domain (Zinc finger domain) of RNF8 (Figure 4B). As an E3 ligase, RNF8 is responsible for transferring ubiquitin bound with E2 to the substrates in the ubiquitin–proteasome system, and the RING domain of RNF8 plays a very important role in it (Figure 4C). 

To determine the role of the RNF8 convergent amino acid (V437I) in the functional convergence of abdominal testicular species, we constructed a mutant RNF8 plasmid by changing Ile into Val at position 437 in RNF8 of the bottlenose dolphin. As RNF8 participates in the regulation of spermatogenesis through the ubiquitination of histones (H2A and H2B), we analyzed the ability of wild-type and mutant RNF8 in bottlenose dolphin to ubiquitinate H2A. We found that the level of substrate H2A protein in the nucleus was significantly reduced 48 h after transfection of the wt-RNF8 and H2A; this meant that H2A was degraded by ubiquitination (Figure 5A,B). Then, the ubiquitination abilities of RNF8 in three groups were compared. Each of the three groups was transfected with Flag-tagged H2A (substrate). At the same time, the control group was transfected with an empty vector (pcDNA3.1); one other group was transfected with wt-RNF8 and another with mut-RNF8. It was found that H2A protein levels in the nucleus of the wt-RNF8 and mut-RNF8 groups were both lower than the control group, and the level of H2A protein in the nucleus of the wt-RNF8 group was significantly lower than that of the mut-RNF8 group (Figure 5C,D). The above showed that the ubiquitination ability of the wt-RNF8 (437I) was stronger than that of the mut-RNF8 (437V).

## 4. Discussion

During embryonic development, if one or both testicles do not fall into the scrotum, known as cryptorchidism, series of risks such as male sterility will arise. According to reports, family inheritance factors, molecular genetic factors and environmental factors may cause cryptorchidism. Analysis of the history of cryptorchidism patients found that about 22.7% of patients had a family history of cryptorchidism [34]; recessive mutation of RXFP2 (c.1496G>A. p. Gly499Glu) can cause familial bilateral cryptorchidism [35]; the G178A polymorphic variant of INSL3 may be linked to cryptorchidism among an Egyptian pediatric cohort [36]. In addition, studies have shown that prenatal exposure to endocrine disruptors (EDCs) may be causally related to congenital cryptorchidism [37]; direct exposure to insecticides in pregnant women has also been shown to have an impact on the prevalence of cryptorchidism [38]. At present, surgical treatment and hormone therapy are the most common treatments for cryptorchidism [39,40].

However, not all mammals have testes that descend into the scrotum, and the testes of many mammals actually remain in the abdominal cavity. It has been reported that testicular descent-related genes (*RXFP2* and *INSL3*) are lost or nonfunctional exclusively in four afrotherians (tenrec, cape golden mole, cape elephant shrew and manatee) that do not undergo testicular descent at all [8]. Thus, we hypothesized that abdominal testicular mammals must have evolved some molecular mechanisms by which the testes adapted to the inner body environment.

Spermatogenesis is a complex process of cell division and differentiation in the seminiferous tubules of the testes. It requires precise expression of structural proteins and enzymes. Our previous studies showed that structural protein genes related to spermatogenesis might have undergone adaptive evolution, and that mammals, mainly laurasiatherians, contain positive selection signals [41]. Recent studies reported that the ubiquitin–proteasome system (UPS) is necessary for various steps of mammalian spermatogenesis, and functions of the spermatogenesis-related UPS genes have been continuously proposed [12,42]. Moreover, evolutionary studies of UPS genes have also been reported, especially in plants and nematodes. For example, it was reported that the ubiquitin-26S proteasome system in Brassicaceae and Poaceae has undergone diversifying evolution [43]; in addition, two large families of ubiquitin-ligase adapters in nematodes and plants have undergone adaptive evolution [44]. However, in mammals, especially abdominal testicular mammals, the evolutionary studies of the spermatogenesis-related UPS genes are unclear. 

In this study, we conducted comparative genomic analyses and functional analyses of the spermatogenesis-related UPS genes involved in regulating the degradation of various proteins in the steps of spermatogenesis using a laurasiatherian dataset, aiming to further explore the molecules mechanisms that abdominal testicular mammals use to maintain healthy reproduction.

We used three different models (free-ratio, two-ratio and branch-site) in PAML to analyze the positive selection of the spermatogenesis-related UPS genes and found that positive selection signals were detected in both the abdominal testicular and scrotal testicular laurasiatherians lineages. However, the positive selection signal detected on the abdominal testicular lineages was far stronger than that detected in the scrotal testicular branches (Appendix A). Moreover, positive selection was specifically detected in six genes (*UCHL3*, *PSMA8*, *USP14*, *MARCH7*, *USP2* and *USP14*) in the abdominal testicular branches (Figure 1 and Appendix A). Notably, when *p* value was corrected by FDR < 0.05, some “signals” showed no statistical significance—e.g., the positive selection signals of *UCHL3* and *PSMA8* specifically detected in the abdominal testicular branches through the branch-site model and positive selection signal of *MARCH7* specifically detected in the abdominal testicular branches through the two-ratio model. This might be because FDR becomes inaccurate due to insufficient sample size [45,46]. In addition, after FDR correction, there were four genes (*UBE2A*, *UCHL3*, *HERC4* and *PSME4*) identified in the abdominal testicular species with a higher ω value than the scrotal species (Figure 2; Appendix A). The six genes in which positive selection was specifically detected on the abdominal testicular branches and four rapid evolution genes all play an important role in spermatogenesis. For example, UCHL3, a deubiquitinating enzyme, participates in regulating germ cell apoptosis and differentiating spermatocytes into sperm cells. UCHL3 knockout mice showed severe testicular atrophy and increased apoptotic germ cells after cryptorchidism injury [25]. In our study, positive selection in *UCHL3* was detected in cetacean lineages (abdominal testicular lineages). Moreover, possible positive selection sites (*p* value tested by LRT < 0.05, posterior probabilities ≥ 0.8) were located in the active center domain of the UCHL3 enzyme (Appendix A). Thus, cetaceans with abdominal testicles might control the loss of sperm cells by strengthening the inhibition of sperm cell apoptosis to deal with “cryptorchidism injury”. 

Convergent evolution has always been a hot spot in the field of evolutionary biology. Different abdominal testicular mammals, although distantly related to one another, have independently evolved similar phenotypes for undescended testes, which could be a typical morphological convergence. To explore whether there is molecular convergence driving this morphological convergence, we performed molecular convergence analysis on 25 spermatogenesis-related UPS genes of laurasiatherians. Eight potential convergent amino acids for eight genes (RNF8 V437I, MARCH7 V314I, UBA7 Q/K713R/E, UCHL3 V64I, PSMA8 N187S, MARCH10 P132S, UBA1 Y323F and RAD18 S440P) were identified. Moreover, convergent amino acids of RNF8 and MARCH7 were verified by representative mammals other than laurasiatherians (Figure 3A–C). For example, for RNF8, most abdominal testicular species were Ile and most scrotal testicular species were Val at position 437 (Figure 3A). However, there were still some abdominal testicular species with the same amino acids as the scrotal testicular species but different from the abdominal testicular species. For instance, at position 437 of RNF8 and position 314 of MARCH7, seals and walruses were not Ile but Val. Studies also reported that seals can use cooled surface blood to directly cool the reproductive organs and protect them from high temperature [10]. This is supported by measurements of testicular temperatures, which have been shown to be lower than “core” temperatures by 6–7 °C in elephant seals [47] and 1–4 °C in harp seals [48]. This anatomical adaptation might explain why the seals did not undergo the above molecular convergence.

Histone ubiquitination is important for nucleosome removal during spermatogenesis and regulating sperm formation. Notably, histone-protamine replacement is affected by temperature. It has reported that histone-protamine replacement and chromatin condensation displayed a significant impairment after 3 months of sauna sessions [49]. Recent studies showed that H2A and H2B ubiquitination regulated by RNF8 was a key step in histone removal [22,50]. RNF8 knockout mice fail to generate mature sperm during spermatogenesis, leading to male sterility [47,51]. Up-regulation of RNF8 can predict the presence of sperm in individuals with azoospermia [52]. The RNF8 is an E3 ligase that contains an N-terminal forkhead-associated (FHA) domain and a C-terminal RING domain [53]. The RING domain of RNF8 can interact with E2 transferring ubiquitin to the substrates (e.g., H2A and HAB) and ubiquitin-tagged substrates can be degraded by the proteasome [54]. Considering that the RNF8 V437I convergent amino acid found in this study was located in the RING domain, RNF8 of abdominal testicular species were expected to display functional convergence. To test this hypothesis, we examined RNF8 function by comparing the ubiquitination ability of two variants and found that the ability of the wild type bottlenose dolphin RNF8 (437I) was stronger than that of the mutant bottlenose dolphin RNF8 (437V), which showed that the RNF8 of abdominal testicular mammals exhibit functional convergence in ubiquitination. This evidence suggests that the mammals with abdominal testes might undergo enhanced histone removal to maintain sperm formation under “cryptorchidism injury” conditions. However, further in vivo experiments, such as site directed mutation of RNF8 in cryptorchidism mouse model, may be needed to support this assertion. 

Overall, the above evidence suggests that spermatogenesis-related UPS genes of abdominal testicular species might have undergone adaptive evolution to stabilize sperm formation in response to “cryptorchidism injury”.

## Figures and Tables

**Figure 1 genes-12-01780-f001:**
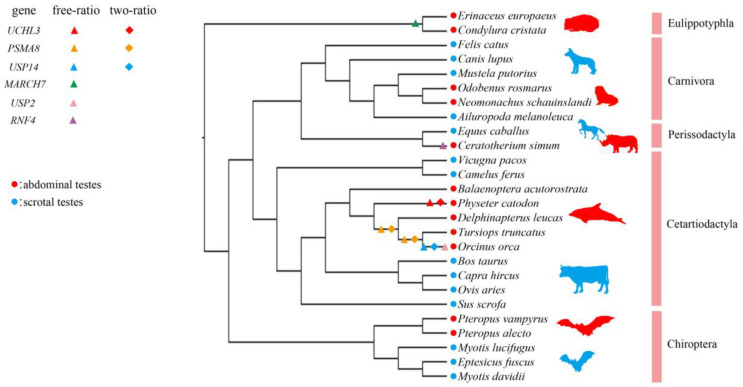
Positively selected genes specific to abdominal testicular branches. Positively selected branches identified by the free-ratio model and two-ratio model are represented by triangles and diamonds, respectively. The six abdominal testicular branch-specific positively selected genes are marked with different colors: UCHL3 (red), PSMA8 (orange), USP14 (blue), MARCH7 (green), USP2 (pink) and RNF4 (purple). The mammals with abdominal testes are in red, and mammals with scrotal testicles are in blue. Adjusted *p* value corrected by FDR < 0.05.

**Figure 2 genes-12-01780-f002:**
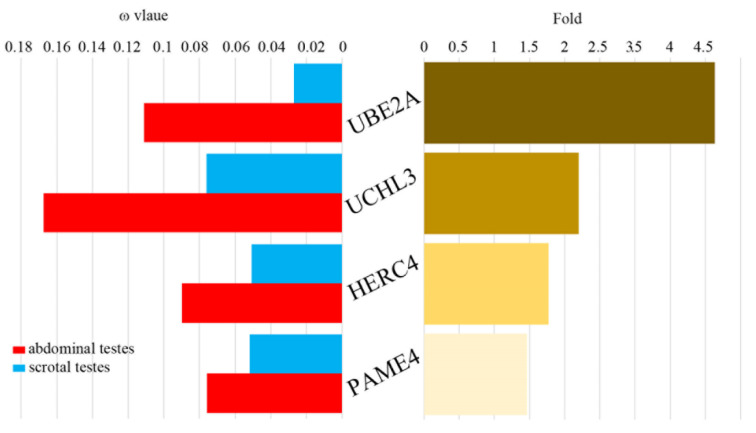
Rapidly evolving genes in abdominal testicular lineages. The nested branch model (two-ratio model) was used to calculate the selection pressure of the foreground branch (abdominal testicular species) and the background branches (scrotal testicular species). The left side shows the ω values of the four accelerated genes for the abdominal testicular and the scrotal testicular species. The ω values of all the genes for the abdominal testicular species are greater than those of the scrotal testicular species. The right side shows the difference between the ω values of the abdominal and the scrotal testicular species for the four genes, sorted according to the difference from high to low. Adjusted *p* value corrected by FDR < 0.05.

**Figure 3 genes-12-01780-f003:**
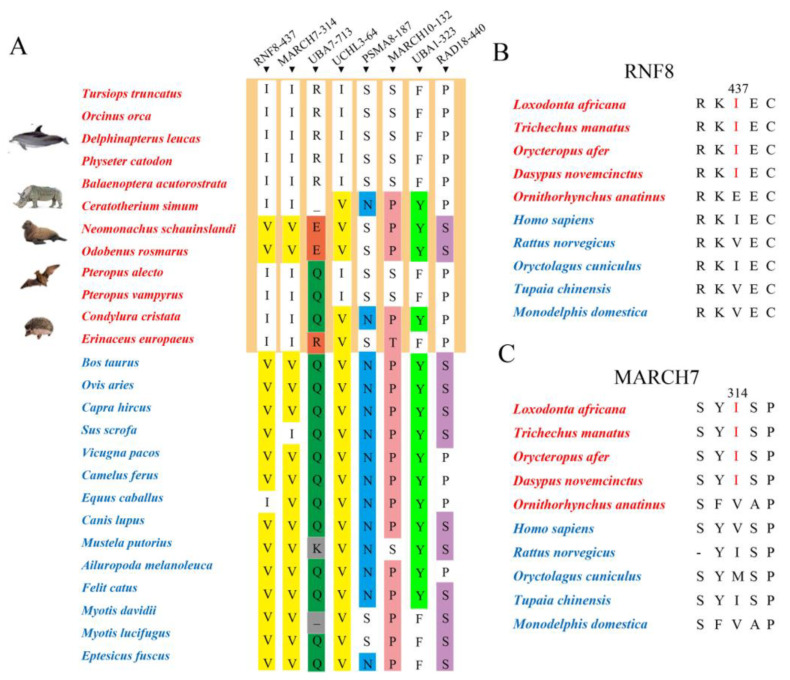
Potential convergent amino acids in species with abdominal testicular species. (**A**) Eight convergent amino acids (RNF8 V437I, MARCH7 V314I, UBA7 Q/K713R/E, UCHL3 V64I, PSMA8 N187S, MARCH10 P132S, UBA1 Y323F and RAD18 S440P) in eight genes identified. (**B**) Amino acid of representative mammalian species besides laurasiatherians at position 437 of RNF8. (**C**) Amino acid of representative mammalian species besides laurasiatherians at position 314 of MARCH7. The mammals with abdominal and scrotal testes are in red and blue, respectively. Yellow: V(Val); Orange: E(Glu)/R(Arg); Green: Q(Gln); Grey: K(Lys)/-; Blue: N(Asn); Pink: P(Pro); Fluorescent green: Y(Tyr); Purple: S(Ser).

**Figure 4 genes-12-01780-f004:**
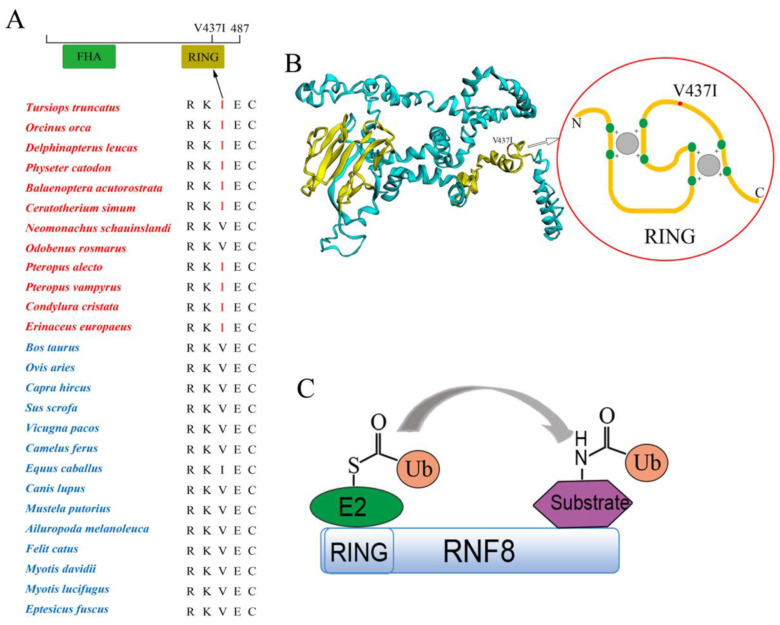
The RNF8 V437I convergence amino acid was identified in species with abdominal testes. (**A**) The convergent amino acid of RNF8 was located at the 437th amino acid of RNF8. (**B**) The V437I convergence amino acid was localized to the RING domain (Zinc finger domain) of RNF8. (**C**) Function schematic of RNF8/RING domain containing V437I convergence amino acid. The mammals with abdominal and scrotal testes are in red and blue, respectively.

**Figure 5 genes-12-01780-f005:**
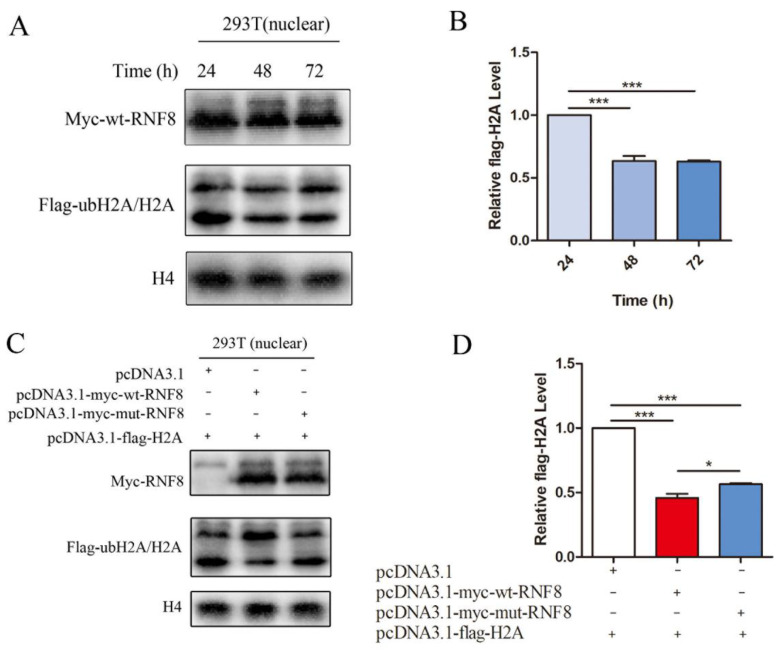
Functional convergence of the RNF8 among abdominal testicular species. (**A**) H2A was clearly ubiquitinated and degraded 48 h after RNF8 was transfected. H2A, RNF8 and H4 proteins were subjected to Western blotting. The blots were probed for H4 as a loading control. (**B**) The blots for H2A proteins. H2A protein level at 24 h was set to 1, and relative H2A protein levels were plotted. (**C**) Bottlenose dolphins of RNF8 ubiquitinated and degraded H2A at a higher level than its mutant. H2A, RNF8 and H4 proteins were subjected to Western blotting. The blots were probed for H4 as a loading control. (**D**) The blots for H2A proteins. When transfected with pcDNA3.1-myc-wt-RNF8 and pcDNA3.1-flag-H2A, H2A protein level was set to 1, and relative H2A protein levels were plotted. Wt-RNF8 stands for RNF8 of bottlenose dolphin (RNF8 437I) and mut-RNF8 stands for mutant RNF8 of bottlenose dolphin (RNF8 437V). Error bars show mean ± SEM (*n* ≥ 3). *p* values were from two-tailed Student’s *t* tests. * *p* < 0.05; *** *p* <0.001.

## Data Availability

Most of the data supporting the findings of this study can be found within the Appendix A; some other data are available from the corresponding author upon reasonable request.

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
