# Peer review of "Insights into the Evolution of Spermatogenesis-Related Ubiquitin–Proteasome System Genes in Abdominal Testicular Laurasiatherians"

_genes, 2021, doi:10.3390/genes12111780_

Round 1
Reviewer 1 Report
In the manuscript "Insights into the evolution of spermatogenesis-related ubiquitin-proteasome system genes in abdominal testicular laurasiatherians", the authors aimed at unravelling the mechanisms supporting ongoing spermatogenesis in abdominal testicular mammals, utilising a total of 26 laurasiatherians representing both abdominal and scrotal testicular mammals. They performed comparative genomic analysis and functional analysis of spermatogenesis-associated UPS genes, including RNF8 and suggested that sperm formation in mammals with abdominal testes is due to increased histone removal and reproductive adaptations.
Overall the manuscript is clear and well written.
Specific comments
- Line 194: there is an extra space between "were then.." and "centrifuged".
- Lines 312-317: there is a slight overlap with introduction.
- At the discussion part, could the authors add a paragraph on any possible clinical relevance of their results, especially for cryptorchid patients (treatment, aetiology)?
- Lines 403-404: could the authors briefly refer to the "in vivo experiments" ?
Author Response
Reviewer #1:
Point 1: Line 194: there is an extra space between "were then." and "centrifuged".
Response 1: Sorry for this mistake, we have removed the extra space.
Point 2: Lines 312-317: there is a slight overlap with introduction.
Response 2: We have cut this passage based on this comment to make it brief.
Point 3: At the discussion part, could the authors add a paragraph on any possible clinical relevance of their results, especially for cryptorchid patients (treatment, aetiology)?
Response 3: According to this comment, we have added a paragraph about the clinical diagnosis and treatment of cryptorchidism in lines 318-328.
Point 4: Lines 403-404: could the authors briefly refer to the "in vivo experiments"?
Response 4: We may explore whether the RNF8 mutation in cryptorchidism mice can save their spermatogenesis. We have added relevant content in lines 420-421.
Reviewer 2 Report
The manuscript (genes-1423277) by Ding et al investigates the underlying molecular mechanisms of testicular descent across numerous species that either have descended testes or not via comparative genomic analyses and some functional assays of the spermatogenesis-related ubiquitin-proteasome system genes essential to sperm formation in laurasiatherians. They identify positive selection and rapid evolution of spermatogenesis-related genes in laurasiatherians without descended testes and ascribe the function of these genes to enhanced histone clearance to stabilize sperm formation to allow the gametes of these animals to not be damaged by their permanent cryptorchidism-like state. The authors then undertake some functional that provided evidence that underpinned their hypothesis . Thus providing some insight into the adaptation of the animals with abdominal testes.
Overall the manuscript is very well written and as far as this reviewer could determine mostly free from typographical/grammatical errors (asides from the minor errors included below as specific points).
The issues encountered in this manuscript that this reviewer could not entirely reconcile:
- Lines 188-202: Could the authors please explain the choice of cell line used for these experiments (ie embryonic kidney vs a more reproductively relevant cell line). Furthermore could this explanation also include why multiple cells lines were not used?
- It is this reviewers option that the discussion could benefit from an extra point surrounding that many of the differences reported in Lines 222-229 are for animals that all seem to be aquatic species – could the potential for exposure to a cold water habitat have driven some of these adaptations?
3 The enhanced histone removal presumably leads to increased protamination – is it this increased % of protamination that protects the sperm form heat damage in the species with abdominal testes? If this is the authors contention – it could added by means of a literature based examination of the protamination % of sperm from these species and added to the manuscript as both data and discussion.
In addition this reviewer has some specific revisions:
- Line 19: The acronym UPS is used without being defined prior, please revise (suggest this acronym could be added to line 17)
- Line 78: The word ‘is’ seems to have been used instead of ‘are’
- Lines 161 & 176-181: It is not clear to this reviewer as to why the focus of these methods is the bottlenose dolphin genes/proteins– please consider adding a rationale as to why this is the case.
Author Response
Reviewer #2:
Point 1: Lines 188-202: Could the authors please explain the choice of cell line used for these experiments (ie embryonic kidney vs a more reproductively relevant cell line). Furthermore could this explanation also include why multiple cells lines were not used?
Response 1: It is reported that 293T cells are used for ubiquitination experiments (Nam Soo Lee et al., 2018; (http://dx.doi.org/10.1074/jbc.M116.765602). Besides, Huey-Wen Shyu et al (http://doi/10.1016/s0014-4827(03)00110-1) also used the 239T cell line when studying the role of RNF36 in spermatogenesis. Coupled with the 293T cell line has the advantages of easier transfection and high protein expression level. Under the trade-off, we chose embryonic kidney cells instead of more
reproductive-related cell lines or multiple cell lines.
Point 2: It is this reviewer option that the discussion could benefit from an extra point surrounding that many of the differences reported in Lines 222-229 are for animals that all seem to be aquatic species – could the potential for exposure to a cold water habitat have driven some of these adaptations?
Response 2: The aquatic species (cetaceans) of Lines 222-229 are secondary aquatic mammals. When their ancestors returned to the sea from land, in order to adapt to aquatic life and reduce energy loss, their bodies were streamlined, and their testicles stayed a high temperature environment of the abdominal cavity while maintaining healthy reproduction. The results show that positive selection signals are mainly concentrated in these aquatic species, which further reveals that they have undergone adaptive evolution in order to maintain healthy reproduction. We think that exposure to cold water habitat could be beneficial to cetaceans to alleviate the high temperature in the abdominal cavity, so it may alleviate rather than drive some of these adaptations.
Point 3: The enhanced histone removal presumably leads to increased protamination – is it this increased % of protamination that protects the sperm form heat damage in the species with abdominal testes? If this is the authors contention – it could added by means of a literature based examination of the protamination % of sperm from these species and added to the manuscript as both data and discussion.
Response 3: We have not found literature reports on the protamine % of sperm from these species with abdominal testes. It may be because abdominal testicular species are mostly protected animals, so the experimental data is less. But the histone-protamine replacement is indeed affected by temperature. Garolla et al. (2013) reported that sauna (high temperature) exposure affects human sperm chromatin structure (histone-protamine replacement and chromatin condensation displayed a significant impairment after 3 months of sauna sessions) and then affect spermatogenesis. We have cited this literature in lines 401-404.
Point 4: Line 19: The acronym UPS is used without being defined prior, please revise (suggest this acronym could be added to line 17)
Response 4: We have defined the acronym UPS in line 4 based on this comment.
Point 5: Line 78: The word ‘is’ seems to have been used instead of ‘are’.
Response 5: Sorry for this mistake, we have revised.
Point 6: Lines 161 & 176-181: It is not clear to this reviewer as to why the focus of these methods is the bottlenose dolphin genes/proteins– please consider adding a
rationale as to why this is the case.
Response 6: Because positive selection signals are mainly concentrated in cetaceans, and bottlenose dolphins is a representative species for studying cetaceans, which have the advantages of better genome annotations and more literature reports. Thus, we chose the genes/proteins of bottlenose dolphins for experiments.
Please see the attachment.
